# Impact of the COVID-19 Pandemic on the Scientific Production of Physical Education Researchers: A Five-Year Follow-Up Study

**DOI:** 10.3390/bs14060475

**Published:** 2024-06-05

**Authors:** Sarah Jane Lemos de Melo, Vanderlei Porto Pinto, Emerson Sebastião, Érica de Moraes Santos Corrêa, Gustavo Christofoletti

**Affiliations:** 1Institute of Health, Federal University of Mato Grosso do Sul (UFMS), Campo Grande 79060-900, MS, Brazil; melosarahjane@gmail.com (S.J.L.d.M.); portovanderley@gmail.com (V.P.P.); 2Department of Health and Kinesiology, University of Illinois Urbana-Champaign, Urbana, IL 61801, USA; esebast2@illinois.edu; 3School of Medicine, Federal University of Mato Grosso do Sul (UFMS), Campo Grande 79060-900, MS, Brazil; ericamscorrea@gmail.com

**Keywords:** COVID-19, bibliometrics, physical education and training, behavioral research

## Abstract

The COVID-19 pandemic caused significant changes in society’s dynamics, particularly affecting the landscape of education. Research in several areas may have been affected during periods of social restrictions. This study analyzed the curricula of 558 researchers across 27 graduate programs in physical education in Brazil to investigate the potential impact of the COVID-19 pandemic on scientific publications. Researchers’ production from 2018 to 2022 underwent a comprehensive analysis, considering the total number of publications, Qualis rank, and journal impact factor. Data were analyzed using chi-squared and Kruskal-Wallis tests. Significance was set at 5%. Overall, the analyzed researchers published a total of 17,932 manuscripts from 2018 to 2022. During the COVID-19 pandemic, there was a decline of 16.4% in the number of articles published (*p* = 0.001). This decline was similar between men and women (*p* = 0.603) and was associated with a worsening in Qualis rank (*p* = 0.001). The number of studies published in journals with impact factors was also affected (*p* = 0.001). The findings suggest a potential impact of the COVID-19 pandemic on the scientific production of Brazilian researchers in the field of physical education. Funding agencies should consider the challenges associated with the COVID-19 pandemic before evaluating researchers and programs.

## 1. Introduction

At the end of 2019, the world was impacted by a new condition responsible for causing fever, dyspnea, pneumonia, and death in individuals of all ages [1]. The first cases, identified in the city of Wuhan, China, showed an alarming rate of dissemination, affecting more than 775 million people in different countries [2]. Histological analyses indicated that the disease resulted from infection with SARS-CoV-2, prompting the World Health Organization to classify it as COVID-19 (referring to the type of virus and the year of disease as identification).

The first case of COVID-19 in Brazil was reported in February 2020 [3]. Similar to other countries, this disease was highly prevalent, with more than 37 million people infected and 700 thousand deaths [2]. In Brazil, this scenario was aggravated by conflicting governmental policies that recommended the use of medications with dubious efficacy against COVID-19 and delayed vaccine acquisition [4,5,6,7].

In early 2020, the World Health Organization declared the novel coronavirus, and the associated disease, COVID-19, a global pandemic. This prompted countries, states, and municipalities to take measures to contain the spread of the virus and reduce the burden on clinics and hospitals. Mask mandates, constant hand hygiene, and shelter-in-place orders were important preventive measures for this disease [8,9]. Social isolation, although necessary to decrease the spread of COVID-19, has affected people’s behavior in situations that require direct human contact [10]. For instance, physical education professionals had to adapt to the pandemic by developing online exercise programs to maintain a minimum level of physical activity for people of all ages [11,12,13].

In situations where social restriction and face-to-face contact were discouraged, research in physical education may have been affected. While the implementation of lockdowns has played an important role in reducing the spread of the virus, hospitalizations, and deaths caused by COVID-19, it also created challenges for conducting new studies requiring direct participant involvement, which potentially affected academic production [14].

The possible impact of the COVID-19 pandemic on scientific production is significant because universities are regularly evaluated by funding agencies based on their published manuscripts. Researchers have reported a reduction in research funding during the COVID-19 pandemic [15,16]. To secure increased financial investment and acquire additional resources, deans exert consistent pressure on researchers to increase research productivity and publish in high-impact journals.

There is growing political pressure on universities to intensify partnerships and enlarge research funding options [17]. The underlying explanation behind the “publish or perish” attitude is that programs with stronger publication records are more likely to attract additional resources. Consequently, scholars who publish infrequently or prioritize activities that do not directly result in publications may find themselves at a disadvantage when competing for academic positions (e.g., assistant professor, lecturer) [18].

In view of the aforementioned, this study investigated the potential impact of the COVID-19 pandemic on the scientific production of graduate programs in Brazil focused on the area of physical education. Confirming the hypothesis that the production of manuscripts was affected by COVID-19, funding agencies should consider the challenges posed by the COVID-19 pandemic before evaluating researchers and programs.

## 2. Methods

This bibliometric study targeted 558 researchers from graduate programs in physical education in Brazil. The study protocol was submitted and approved by the Internal Review Board (protocol: # 5.454.817). The study was conducted in accordance with the Guideline for Reporting Bibliometric Review of the Biomedical Literature (BIBLIO) [19].

The inclusion criteria included researchers who were faculty members of graduate programs in physical education. The exclusion criteria involved researchers who were members of programs created after 2017 because, in Brazil, a master’s thesis takes up to two years to be completed and a doctorate dissertation takes up to four years; therefore, researchers of programs created after 2017 would not have their scientific production assessed before the COVID-19 pandemic. Out of the 39 graduate programs in physical education in Brazil, 27 met the eligibility criteria.

After the selection of the graduate programs, all faculty members were cataloged. The scientific production from 2018 to 2022 of the researchers was analyzed. The definition for this period was based on the COVID-19 pandemic in Brazil. Since the first case of the disease was identified in February 2020 [3], analyzing the years 2018 and 2019 would allow us to observe the scientific production before the COVID-19 pandemic. In 2020 and 2021, the disease was highly prevalent, and lockdowns and shelter-in-place (social restrictions) measures were implemented [20]. In 2022, the disease was still present, but was now mitigated by extensive vaccination of the Brazilian population [21].

Researchers’ production was limited to the publication of peer-reviewed articles published in scientific journals. The other forms of scholarly work such as abstracts, event presentations, pre-prints, thesis, and dissertations were not considered in the analysis. Despite being important academic work, they are usually not evaluated by funding agencies.

The scientific production of the researchers was extracted from the Brazilian Curriculum Lattes platform (https://lattes.cnpq.br/, accessed on 1 March 2024). Briefly, the Lattes Platform is an information system maintained by the Brazilian federal government to manage information on science, technology, and innovation related to research in Brazil. This database is a digital curriculum vitae/memorial in which researchers document their scientific production [22]. By creating and submitting records to Curriculum Lattes, researchers acknowledge that this information is available to the public and authorize its use to support the evaluation of graduate programs and research. To ensure data privacy, the names of the researchers were preserved and only general information about the studies (such as year of publication and journal indexing) was extracted. In this study, the scientific production of all researchers was analyzed from 2018 to 2022 (5-year follow-up). Analyses were conducted between March and December of 2023.

The scientific production of the researchers was evaluated using the number of manuscripts published annually, irrespective of the quality of the studies. Additionally, two specific metrics known to assess the quality and representativeness of journals in the scientific community were adopted: the Qualis rank system and the Web of Science^®^ impact factor [23,24].

Qualis is the official journal ranking system of the Brazilian government and is managed by the CAPES Foundation (Portuguese: Coordenação de Aperfeiçoamento de Pessoal de Nível Superior) [25]. This classification system assigns grades to journals based on their circulation level (i.e., local, national, or international). The current Qualis rank system categorizes journals into three main groups: “A” (this includes subcategories A1, A2, A3, and A4), “B” (this includes subcategories B1, B2, B3, and B4), and “C”. Journals in the “A” category have the highest representability in the scientific community, typically featuring international circulation and indexed in important databases such as PUBMed^®^. Journals in the “B” category have good representability in the scientific community, generally with national circulation, and are indexed in databases such as SciELO^®^. Journals in the “C” category usually have a local circulation.

The impact factor of the journals was assessed using the Web of Science^®^ Journal Citation Reports (JCR). The JCR is a comprehensive resource that provides data on academic journals in the sciences and social sciences. The journal’s impact factor—one of the most well-known metrics provided by the JCR—is used to measure the importance of journals by calculating the number of times articles have been cited in recent years [26]. The higher the journal’s impact factor, the better the journal’s classification.

### Statistical Analysis

Data were presented as absolute frequencies, means, and standard deviations. Inferential analyses were performed using non-parametric tests, as these are normally applied to ordinal and nominal data. The chi-squared test was used to analyze the variability in the total number of articles published before and during the COVID-19 pandemic. This test, along with likelihood ratio analysis, was used to assess the impact of the COVID-19 pandemic according to the region of the graduate program in Brazil, the researcher’s sex, and the Qualis rank system. Additionally, the Kruskall-Wallis *H* test was used to analyze the impact factor of the journals in which the articles were published, with the year of publication being included as an independent variable and the impact factor as a dependent variable. For all analyses, significance was set at 5%.

## 3. Results

Twenty-seven graduate programs in physical education were analyzed in this study. The programs are located in different regions of Brazil: twelve are located in the Southeast, seven in the South, six in the Northeast, and two in the Midwest region of the country. There was no graduate program in physical education in the North region of Brazil that met the eligibility criteria. A total of 558 researchers were included. The majority were male (70.8%, *p* = 0.001), with a significant portion of faculty members affiliated with programs located in the Southeast region (48.9%, *p* = 0.001). Figure 1 provides detailed characteristics of the programs and researchers included in this study.

A total of 19,296 manuscripts were extracted and analyzed from the Curriculum Lattes of the researchers. Out of the 19,296 articles, 1364 were excluded due to repeated publications, which occurred when more than one researcher was involved in the same study, causing the article to be counted multiple times. Between 2018 and 2022, 17,932 studies were published. The impact of the COVID-19 pandemic on the scientific production of researchers was observed between 2021 and 2022, with a 16.4% decline in the number of articles published. This decline was observed in all regions of Brazil and was similar between men and women. Table 1 displays the number of articles published from 2018 to 2022, separated by region and sex.

The COVID-19 pandemic negatively impacted the Qualis ranks of the manuscripts published. Similar to the total number of studies, the impact on the Qualis ranks occurred between 2021 and 2022, with a 14.7% decline in the publication of articles in journals categorized as “A”, 21.6% as “B”, and 11.1% in the “C” category. Table 2 lists the number of articles published separated by the Qualis rank system.

Of the 17,932 manuscripts analyzed, 9914 (55.29%) were published in journals with an impact factor. The number of studies published in journals with impact factors was affected by COVID-19, particularly between 2021 and 2022. By contrast, the mean score of the impact factor of the journals was not affected. Figure 2 shows the number of articles published in journals with JCR impact factors and Table 3 shows the mean impact factor scores for each year.

## 4. Discussion

This study investigated the impact of the COVID-19 pandemic on the scientific production of Brazilian researchers who are members of graduate programs in physical education. The results indicated a perceptive decline in the production of manuscripts, particularly between 2021 and 2022. The decline was similar between men and women. The Qualis rank was affected by COVID-19, resulting in a decline in publications in journals categorized as “A”, “B”, and “C”. The quantity of articles published in journals with an impact factor was also impacted. These findings confirm our hypothesis that the scientific production and associated metrics of evaluation (i.e., quantity, journal’s impact factor, and Qualis rank) would be negatively impacted by the COVID-19 pandemic. Understanding these parameters is important for analyzing the research scenario during the COVID-19 pandemic and the challenges faced by researchers and programs. This has further practical applications regarding the evaluation of researchers and graduate programs by funding agencies.

Most graduate programs in Brazil are located in the Southeast region of the country. The Southeast part of Brazil has some of the largest cities, such as São Paulo, Rio de Janeiro, and Belo Horizonte. Consequently, a significant number of researchers are in this region. This characteristic explains the data presented in Table 1, which demonstrates a higher quantity of manuscript publications in programs from this region compared to other regions.

The impact of the COVID-19 pandemic on scientific production has occurred in all regions, with a decline observed between 2021 and 2022. Considering that the first case of COVID-19 in Brazil occurred in 2020, a decline in scientific production from that year onward would be expected. However, from data collection to its publication, there is usually a time-lapse of 1–2 years [27,28]. In other words, the increase in production in 2020 and 2021 was the result of studies conducted and developed between 2018 and 2019, before the COVID-19 pandemic. Conversely, the decline in production in 2022 reflects the impact of the COVID-19 pandemic, as it encompasses articles developed between 2020 and 2021. To this end, the impact of the pandemic on the research is evident in the publications of 2022.

The number of articles published by men was greater than the number of articles published by women. These data reflect the historical profile of universities and science, which typically employ more men than women [29,30]. Considering that women represent only 29.2% of the total researchers in physical education programs in Brazil, their scientific production reaches a proportion close to this analysis (4401 articles, representing 24.5% of the production in the area). The decline in scientific production was similar between men and women. We expected to find a greater decline in the scientific production of women considering that, with the lockdown imposed during the pandemic, women were burdened with household activities and childcare in addition to their responsibilities as researchers [31]. This finding reinforces women’s efforts to mitigate the impact of the COVID-19 pandemic on their scientific production, resulting in a decline similar to that observed for men.

The Qualis rank system is a method used to assess the scientific production of researchers and programs in Brazil [32]. It considers both the indexing and the Scopus quartile ranking of the journals. As these metrics are constantly changing; the Qualis rank system is, from time to time, updated by the Brazilian government. There have been many criticisms regarding Qualis as a mechanism for measuring quality. It has been argued that using a single metric to evaluate different areas of expertise tends to cause distortions in criteria [33]. In addition, journals classified in the lower categories of Qualis tend to remain in the same category because researchers do not submit their manuscripts to these journals. This prevents journals from improving their Qualis rank [34]. Despite the criticism of Qualis, we chose to use this metric because it is a federal governmental mechanism for evaluating graduate programs in Brazil.

When analyzing the Qualis rank, the total number of articles published in journals categorized as “A”, “B”, and “C” declined during 2021 and 2022. Considering that categories “A” and “B” denote journals of higher quality, this finding allows us to assert that not only was the quantity of manuscripts impacted by the COVID-19 pandemic, but also the quality of the studies developed.

Similar to Qualis, there are many criticisms regarding the use of impact factors to evaluate the quality of an article [35]. If a journal is evaluated based on the number of citations, there may be a bias caused by researchers and reviewers who forcibly cite and recommend their references, seeking to improve the impact factor of the journals and their chances of publication [36]. Since there is no consensus on the best form to evaluate the quality of research, we chose to include an analysis of the impact factor, along with Qualis.

Figure 2 shows that the number of articles published in journals with impact factors declined between 2021 and 2022. This finding reinforces that the impact of COVID-19 on Brazilian researchers occurred in 2022. The impact factors of the journals in which the studies were published remained consistent over the years (Table 3). This demonstrates an effort of researchers to maintain the quality of their scientific production, even though the number of articles published in journals with impact factors has declined.

For the screening of scientific production, it is possible to use the Curriculum Lattes platform or perform an active search in databases such as PubMed^®^, Scopus^®^, or EMBASE^®^. We opted to use Curriculum Lattes for three reasons: First, many journals are present in more than one database, which could lead to the inclusion of repetitive articles. Second, some journals do not provide an option to search for recently accepted articles; on the other hand, the Curriculum Lattes platform allows researchers to enter information about an approved article immediately after receiving an acceptance letter from the journal. Third, Curriculum Lattes is a tool that each researcher must update regularly as programs are constantly evaluated [37]. For these reasons, we chose to conduct an analysis on the researchers’ Curriculum Lattes rather than on the databases.

Our findings confirm the results of previous studies on the impact of the pandemic on scientific research and clinical academic training [38,39,40]. According to Riccaboni and Verginer [39], the pandemic induced a sudden increase in research output related to COVID-19 and a significant drop in overall publishing rates in areas unrelated to COVID-19. Raynaud et al. [40] identified an 18% decrease in non-COVID-19 research during the pandemic. This finding is consistent with those of the present study. However, it is important to highlight that our study included only researchers working in the field of physical education, whereas Raynaud et al. [40] conducted a broader study without a focus on any specific field.

This study had some limitations. First, the exclusion of repeated manuscripts identified in Curriculum Lattes was based on the first appearance of the study in the statistical spreadsheet and did not consider the order of authorship or the research location. This might impact the number of articles developed by men or women or the region in which the study was produced. Second, it is possible that a researcher is affiliated with more than one graduate program, and their production may pertain to two distinct areas. In these cases, all the articles were counted because it was not possible to precisely identify the program to which each study belonged. Third, we standardized the use of metrics (Qualis and impact factor) of journals in 2023. It is possible that a journal had a change in Qualis or impact factor score during the 5-year timeline of this study [41]. Despite these limitations, we were able to provide valuable information on the quantity and quality of Brazilian researchers’ scientific production and how it was impacted by the COVID-19 pandemic. Future studies should attempt to address the potential impact of the COVID-19 pandemic in other areas to determine if the pattern observed in physical education holds true in different fields. Since COVID-19 has waned, future studies should further investigate whether such publication metrics have returned to levels prior to the pandemic.

## 5. Conclusions

The COVID-19 pandemic negatively affected the scientific production of physical education researchers in Brazil. Both the quantity and quality of manuscripts were impacted with no clear distinction of such impact in the scientific production of both men and women. The results should be considered by government agencies when evaluating graduate programs in the area of physical education—although other areas have most likely been impacted as well—during the pandemic. This is important because although COVID-19 has waned, some of its consequences may still linger in different areas of society, including, but not limited to, scientific research and its protagonists: researchers and participants.

## Figures and Tables

**Figure 1 behavsci-14-00475-f001:**
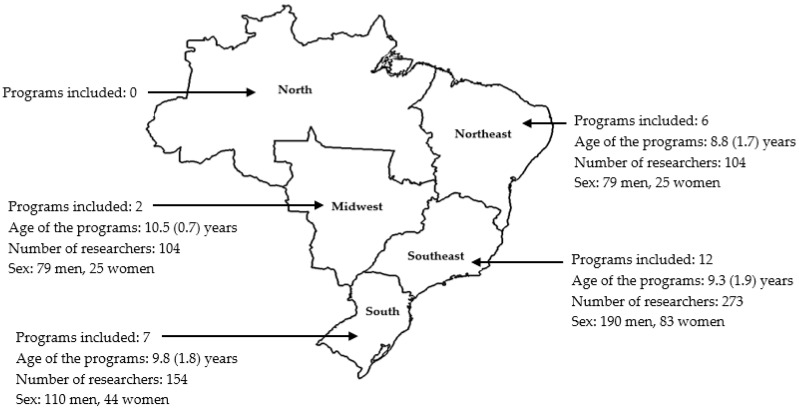
Characteristics of the programs and researchers in Physical Education in Brazil.

**Figure 2 behavsci-14-00475-f002:**
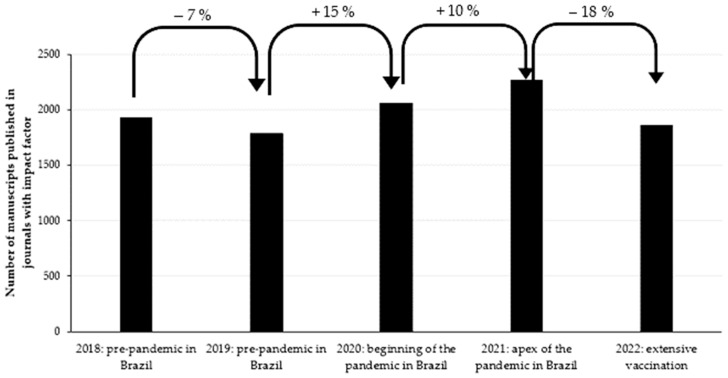
Impact of COVID-19 on the number of studies published in journals with impact factors.

**Table 1 behavsci-14-00475-t001:** Impact of COVID-19 on the number of studies.

Variables		Year of Publication					Total	*p*
		2018	2019	2020	2021	2022		
Number of studies, n		3493	3311	3840	3969	3319	17,932	0.001
Number of studies by region, n	South	1169	1038	1153	1191	943	5489	0.001
	Southeast	1638	1523	1714	1747	1577	8199	
	Midwest	158	193	203	240	186	980	
	Northeast	528	557	770	791	613	3259	
Number of studies by sex, n	Male	2660	2468	2898	3008	2497	13,531	0.603
	Female	833	843	942	961	822	4401	

**Table 2 behavsci-14-00475-t002:** Impact of COVID-19, according to the Qualis rank.

Qualis Rank	Year of Publication					Total	*p*
	2018	2019	2020	2021	2022		
A1	473	437	578	617	545	2650	0.001
A2	402	390	445	483	438	2158	
A3	263	241	293	281	220	1298	
A4	292	248	297	299	230	1366	
B1	843	746	752	638	619	3598	
B2	579	589	553	508	321	2550	
B3	125	147	198	206	121	797	
B4	71	89	88	70	54	372	
C	445	424	636	867	771	3143	
Total	3493	3311	3840	3969	3319	17,932	

**Table 3 behavsci-14-00475-t003:** Impact of COVID-19, according to the Web of Science^®^ Journal Citation Report (JCR).

	Year of Publication					*p*
	2018	2019	2020	2021	2022	
Mean JCR score (95% Confidence Interval)	2.501 (1.9; 3.0)	2.157 (1.8; 2.4)	3.256 (2.5; 3.9)	2.883 (2.3; 3.4)	1.932 (1.8; 2.0)	0.275

## Data Availability

The data presented in this study are available on request from the corresponding author.

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
