# Peer review of "Impact of the COVID-19 Pandemic on the Scientific Production of Physical Education Researchers: A Five-Year Follow-Up Study"

_behavsci, 2024, doi:10.3390/bs14060475_

Round 1

Reviewer 1 Report

Comments and Suggestions for Authors

Congratulations on your work. The paper is generally interesting and well presented, but there are some issues I would like you to take into account for a future revision:

- Reference 10 is a self-reference for which I am sure you can find dozens of publications that say the same thing.
The approval from the Internal Review Board is mentioned, but more details on ethical considerations and data privacy could be included.
-
Including more detailed analyses or visual aids, such as graphs or charts, could enhance the presentation of the results.
-
More discussion on the variability within regions and between different types of publications could provide deeper insights.
-
Suggestions for future research directions could be expanded to provide a roadmap for subsequent studies. Discussing broader implications for other fields of research and generalizing the findings could add value to your work.

Reviewer 2 Report

Comments and Suggestions for Authors

Comments on the Quality of English Language

This paper was well-written with few minor spelling and grammatical issues that need addressing.

Round 2

Reviewer 2 Report

Comments and Suggestions for Authors

Thank you for your responses to the reviewers' comments and the attention to detail in your manuscript revisions. The requests have been sufficiently satisfied (or sufficient justifications have been provided for rejected revision requests). 

Comments on the Quality of English Language

The writing is clear and uses appropriate language for the variables of interest, methods used, and interpretations drawn.